# The Detrimental Role of Intraluminal Thrombus Outweighs Protective Advantage in Abdominal Aortic Aneurysm Pathogenesis: The Implications for the Anti-Platelet Therapy

**DOI:** 10.3390/biom12070942

**Published:** 2022-07-05

**Authors:** Xiaoying Ma, Shibo Xia, Guangqin Liu, Chao Song

**Affiliations:** 1State Key Laboratory of Bioreactor Engineering & Shanghai Key Laboratory of New Drug Design, School of Pharmacy, East China University of Science and Technology, Shanghai 200237, China; y20150160@mail.ecust.edu.cn; 2Keenan Research Centre for Biomedical Science, St. Michael’s Hospital, Toronto, ON M5B 1W8, Canada; 3Department of Vascular Surgery, Shanghai Changhai Hospital, Shanghai 200433, China; 15000508808@163.com (S.X.); xxliuyedao@163.com (G.L.)

**Keywords:** abdominal aortic aneurysms, intraluminal thrombus, vascular biology, anti-platelet drugs, aspirin

## Abstract

Abdominal aortic aneurysm (AAA) is a common cardiovascular disease resulting in morbidity and mortality in older adults due to rupture. Currently, AAA treatment relies entirely on invasive surgical treatments, including open repair and endovascular, which carry risks for small aneurysms (diameter < 55 mm). There is an increasing need for the development of pharmacological intervention for early AAA. Over the last decade, it has been increasingly recognized that intraluminal thrombus (ILT) is involved in the growth, remodeling, and rupture of AAA. ILT has been described as having both biomechanically protective and biochemically destructive properties. Platelets are the second most abundant cells in blood circulation and play an integral role in the formation, expansion, and proteolytic activity of ILT. However, the role of platelets in the ILT-potentiated AAA progression/rupture remains unclear. Researchers are seeking pharmaceutical treatment strategies (e.g., anti-thrombotic/anti-platelet therapies) to prevent ILT formation or expansion in early AAA. In this review, we mainly focus on the following: (a) the formation/deposition of ILT in the progression of AAA; (b) the dual role of ILT in the progression of AAA (protective or detrimental); (c) the function of platelet activity in ILT formation; (d) the application of anti-platelet drugs in AAA. Herein, we present challenges and future work, which may motivate researchers to better explain the potential role of ILT in the pathogenesis of AAA and develop anti-platelet drugs for early AAA.

## 1. Introduction

Abdominal aortic aneurysm (AAA) is a common cardiovascular disease, usually caused by smoking, genetic variation, and atherosclerosis, with the risk of rupture [1]. In most cases, AAA has an asymptomatic pathology. Ruptured aneurysms lead to death in 30–50% of cases. To prevent rupture risk and other risk factors, aneurysms larger than 5.5 cm in diameter require endovascular or open repair [2,3]. However, some small AAAs grow rapidly and may rupture before they reach 5.5 cm in diameter [4]. In addition, AAAs with the same maximum diameter also showed different growth rates [5]. AAA growth rate is one of the most important factors directly related to rupture risk. The data indicate that AAA grows at an average rate of 2.5 to 7.9 mm per year [6]. In addition, AAA diameter and peak aortic wall stress are also classic parameters for rupture risk assessment.

In the AAA vessel wall, infiltration of inflammatory leukocytes, neovascularization, and activation of proteolytic enzymes are involved in extracellular matrix degradation, ultimately resulting in a weakened vessel wall [7,8]. Based on the above findings, the deposition of intraluminal thrombus (ILT) plays an important role in the progression of AAA, as the thrombus can modulate wall stress [9] and inflammatory molecules and cells in the vessel wall [10].

The ILT has a complex fibrin structure consisting mainly of canaliculi, platelets, erythrocytes, and other hematopoietic cells [11]. Clinically, ILT is present in 75% of AAA sacs [12]. Mural thrombus is associated with arterial wall hypoxia [13,14], cellular inflammation, and extracellular matrix apoptosis [10], and can promote aneurysm growth and ultimately lead to aneurysm rupture [15]. Conversely, ILT is also thought to reduce the risk of rupture because its attachment reduces the impact of blood flow shear stress on the vessel wall [9]. However, although ILT plays an important role in the progression/rupture of AAA (both protective and detrimental), the mechanism of ILT formation and deposition has not been fully elucidated.

## 2. The Formation and Deposition of ILT in AAA

Multiple studies have shown that the thickness of the ILT correlates with its rapid expansion [6,16]. Thin ILTs, however, allow inflammatory cells to penetrate deeper into the ILTs more efficiently and act on the vessel wall. In addition, in thin ILTs, proteins produced in ILTs or absorbed from circulating blood cling to the vessel wall, so it appears that thinner ILTs are more damaging to the vessel wall than thicker ILTs [11,17]. Thus, we need to summarize the spatiotemporal distribution of ILT and layer-specific variability to globally analyze the impact of ILT on AAA progression/rupture [18].

### 2.1. ILT Structure (Biochemical Perspective)

Most ILTs spatially have three layers, known as luminal, medial, and abluminal. There is a clear demarcation and weak adhesion between two adjacent layers. Some specific cells and molecules, such as platelets, macrophages, P-selectin, MMPs, Fibrin, etc., affect the formation and deposition of the ILT.

Over time, discrete ILTs mature and form channels called canaliculi. These canaliculi connect the lumen to the lumen layer, allowing various types of cells in the blood vessel to penetrate into the interior of the ILT from all directions for material exchange [11]. The cell types present in the luminal layer canaliculi are usually degranulated platelets and macrophages. There is a biological balance of coagulation and lysis at the luminal interface between the ILT and circulating blood. Therefore, the lumen of the AAA is rarely obstructed. The physiological activity present in the luminal interface in contact with circulating blood involves platelet activation, which releases microparticles and exposes phospholipids [19]. The involvement of platelets is elucidated in Section 4.3: The role of platelets in ILT formation.

Macrophages do not appear to be passively trapped in ILT, as they do not display a necrotic or apoptotic phenotype [11]. A unique subset of activated macrophages was assembled within the luminal layer [20]. These macrophages secrete various anti-inflammatory cytokines, and these macrophage subtypes are distinct from the macrophage subtypes in adventitia that produce nitric oxide and reactive oxygen species (ROS) intermediates [21]. However, the exact contribution of this distinct macrophage subset to the formation of ILTs remains unclear. In the future, we need further studies to determine the distribution and roles of different subtypes of macrophages within the ILT.

P-selectin is an adhesion molecule expressed on the surface of activated cells. At the luminal interface between the ILT and circulating blood, platelet aggregation promotes neutrophil adhesion and activation, a biological process mediated primarily by P-selectin expression [22]. In the most luminal layer of the ILT, activation of neutrophils forms a neutrophil extracellular trap (NET) that traps many enzymes, including proteases, MMP-9, elastase, and pro-oxidases, which are subsequently gradually released. The roles of P-selectin and neutrophil are elucidated in Section 4.3: The role of platelets in ILT formation and Section 4.4: Platelet-related hemostatic proteins.

Platelets [23] and macrophages [24] are important sources of matrix metalloproteinases (MMPs), which can proteolytically damage aortic structures. In addition, platelet-derived chemokines can modulate the expression of MMPs from vascular smooth muscle cells (VSMCs) and macrophages [25]. It should be noted that despite the presence of a large number of proteases in the heterogeneous ILT structure, the protease activity mainly resides in the luminal layer [26]. In contrast, the proteases of the abluminal layer were mostly inactive, possibly due to the presence of excess protease inhibitors. Therefore, the balance of protease and its inhibitor expression plays an important role in stabilizing the integrity of the AAA arterial wall.

Some authors suggest that the ILT structure contains large, dense, cell-free fibrin-rich regions that reduce fibrinolysis and stabilize thrombus volume. Fibrin is uniformly deposited throughout the ILT [27]. Fibrin formation results in the retention of plasminogen and tissue plasminogen activator (t-PA), both of which are highly capable of binding fibrin polymers. In AAA, initial retention of plasminogen and t-PA in plasma and urokinase plasminogen activator (uPA) and its inhibitors in neutrophils occurs in the outermost side of the luminal layer near the lumen [28]. Activation of plasmin promotes fibrinolysis, which releases fibrin degradation products. The strongest fibrinolytic activity is at the abluminal interface between the ILT and the wall [29]. In addition to its proteolytic ability, plasmin activates MMPs, mobilizes TGF-β, and degrades adherent pericellular proteins such as fibronectin (Fn), ultimately inducing mesenchymal cell detachment and death [30]. The number of neutrophils in ILT is more than ten times higher than in circulating blood because these cells have high affinity for the fibrin–fibronectin network via integrins [31], and bind to P-selectin on platelets [32].

Fn, a dimer of a 250 kDa subunit, is a key component of the extracellular matrix. Fn mainly exists in two forms, known as plasma Fn (pFn) and cellular Fn. Studies have found that pFn can promote platelet aggregation when bound to fibrin, but inhibit platelet aggregation when fibrin is absent [33]. Therefore, we hypothesized that pFn may be endowed with dual functions in AAA (depending on the specific microenvironment); pFn can either support or inhibit ILT formation. We recommend further studies to explore the role of changes in fibrin structure and pFn content in ILT formation and AAA progression.

Based on the above findings, we can conclude that the effect of ILT on the arterial wall is dynamic as ILT grows. Corresponding interventions for different stages of ILT have clinical prospects.

### 2.2. Hemodynamics in ILT Formation

Hemodynamic parameters characterizing blood flow in the AAA lumen may play a key role in initiating and maintaining some of the cascades that promote the formation and deposition of ILTs.

On the one hand, the presence of ILT reduces the mechanical stress of blood flow on the aneurysm wall, thus reducing the risk of rupture [9]; on the other hand, the deposition of ILT gradually weakens the strength of the arterial wall [14]. The formation of the thrombus layer is related to the local blood flow microenvironment within the AAA, but the specific mechanisms leading to thrombosis remain incompletely understood. Regarding the prediction of ILT deposition, the researchers shifted their focus from mechanical stresses inside the vessel wall to hemodynamical shear stresses on the lumen wall [34]. Colciago et al. demonstrated that wall shear stresses and ILT deposition are linked through numerical solutions of mathematical flow models; in addition, time-averaged wall shear stress (TAWSS) can be used as an indicator for preliminary assessment of rupture risk in ILT-deposited areas [35].

It has been reported that vortical structures may play an important role in regulating ILT deposition in specific wall regions. The low TAWSS will be modulating the growth of thrombus within these preferred ILT accumulated regions. Biochemically, decreased WSS values correlate with decreased production of prostacyclin, prostaglandins [36], and nitric oxide (NO) [37], which are involved in a variety of biological signaling and protect arterial walls from the damaging effects of prothrombotic activities. Thus, as several studies have pointed out, low levels of TAWSS represent a loss of antithrombotic protection, which may lead to the adhesion of prothrombotic agonists (e.g., platelets and leukocytes) [38,39,40].

From the perspective of hydrodynamics, the larger the aneurysm volume, the lower the wall shear stress, and the more obvious the local vortex formation, which promotes the formation of ILT. Besides, for small abdominal aortic aneurysms, the hemodynamic changes within the aneurysm are small, which allows other non-hemodynamic factors to influence the formation of ILT.

## 3. The Detrimental Role of ILT in the Progression of AAA

### 3.1. ILT Presence, Thickness, and Volume

The presence of thin circumferential thrombus is seen as a risk factor for increased aneurysm growth rate and risk of rupture, and may also indicate the timing of surgery [41]. Domonkos et al. revealed that thicker thrombi slowed AAA growth and vice versa; thinner ILTs were associated with higher AAA growth rates [42]. However, when the ILT reaches a certain thickness, it causes the AAA wall to become fragile due to hypoxia [14].

Furthermore, studies have shown that the thickness of ILT may affect aneurysm wall remodeling [43] and may also contribute to the generation of oxidative stress and its spatial distribution in the aneurysm. A recent study demonstrated significantly higher levels of oxidative stress and proteolytic enzymes within aneurysm walls covered by thin thrombi (≤10 mm) compared with those covered by thick thrombi (≥25 mm) [41].

Besides, ILT thickness was associated with apoptosis of VSMCs and positively correlated with MMP-2 concentrations in the underlying wall [16]. Vorp et al. [14] claimed that thicker ILT promotes local hypoxia, promotes inflammation of the underlying aneurysm wall, and attenuates tensile wall strength resulting from elastic fiber degradation. ILT provides an ideal microenvironment for the generation and retention of proteolytic agents [18]. These chemicals can be transported through canaliculi in ILT [44] and degrade elastin and collagen through proteolytic pathways, thereby weakening wall strength. This wall weakening mechanism could explain why the presence of ILT is associated with aneurysm growth rate [45] and why a thick ILT layer [10] and rapid increase in ILT volume [46] have been linked to AAA rupture risk. As mentioned earlier, larger thrombi can lead to higher rates of AAA growth, which can weaken the aneurysm wall [47]. For example, Kazi et al. [10] revealed that aneurysms covered by thrombus had thinner walls and exhibited inflammatory cell infiltration, apoptosis of VSMCs, and extracellular matrix degradation. This means that thrombosis and the accumulation of inflammatory cells may disrupt the structural integrity and stability of the vessel wall, thereby increasing the risk of aneurysm rupture. ILT induces an inflammatory environment in which neutrophils, cytokines, proteases, and reactive oxygen species are sequestered, which reduces wall strength. ILT thickening is associated with increased levels of MMPs, elastin degradation, apoptosis of aortic wall smooth muscle cells, and local tissue hypoxia. Satta et al. [48] suggested that thrombus thickness was associated with rupture risk; ruptured AAAs had thicker ILTs compared to unruptured AAAs. Wolf et al. [6] proposed that large intraluminal thrombus load, as determined by mean TARC and thrombus volume fraction, was positively associated with aneurysm expansion rate, whereas initial aneurysm size was not associated with aneurysm expansion rate. Hans et al. [49] found that ruptured AAAs were larger in diameter and had larger thrombus volumes than intact AAAs. Stenbaek et al. [46] demonstrated that the presence of ILT is associated with an increased risk of rupture in small AAAs (AAA diameter ≤ 5 cm). Furthermore, in ruptured AAA, ILT had a higher growth rate. Therefore, for small AAAs, the rapid increase in intraluminal thrombus burden should be used as a new gold standard for early surgical repair, rather than relying on the diameter of the AAA to determine the timing of surgical repair. One limitation common to all of the above studies was that patients with ruptured and intact AAAs were not matched for maximum AAA diameter. Therefore, we need to investigate the effects of parameters such as volume/thickness/growth rate of ILT on rupture in AAAs of the same diameter.

Our clinical findings support the idea that the protective biomechanical advantage ILT provides via lowering wall stress is outweighed by weakening of the AAA wall (as shown in Figure 1), particularly in patients with small rAAAs (rupture AAAs), which is consistent with one study suggesting that increased thrombus burden (normalized thickness and percentage ILT volume) is associated with AAA rupture at smaller diameters [50].

However, there are also studies reporting that ILT has no effect on aneurysm wall rupture. Schurink et al. [51], who measured intrathrombus pressure during open AAA repair, proposed that ILT within the aneurysm does not reduce pressure near the aneurysm wall and, in turn, does not reduce the risk of rupture. The results of this study are consistent with Fillinger’s clinical findings [52]; abdominal 2-D CT scans of 259 patients with AAA (122 ruptured, 137 elective) showed that the maximum thrombus thickness and thrombus circumference did not differ between ruptured and unruptured AAAs. Golledge et al. also showed that there was no significant difference in ILT volume between ruptured and ruptured AAAs with matched diameters [53].

The above findings suggest that whether ILT thickness and volume can be used to assess rupture risk remains controversial, as the role of ILT in AAA progression and rupture (protective or detrimental) depends on multiple factors (mechanics and molecular biology; some still unclear or unknown).

### 3.2. ILT Spatial Distribution

Although there have been many publications on the effect of ILT on rupture, most have focused on ILT size-related characteristics such as volume/area growth [12,53], relative sac volume, and maximum thickness [48]. There are few reports on the effect of ILT spatial distribution on rupture [49]. So, does the spatial distribution of ILT affect AAA progression/rupture? Most AAAs exhibited asymmetric ILT deposition favoring the anterior side [49]. Different deposition patterns of ILT may have different effects on AAA progression/rupture.

Recently, Metaxa et al. [54] found that AAA with posterior thrombus deposition was associated with significantly lower growth rates and posterior maximum wall stress compared to AAA with anterior thrombus deposition. Therefore, posterior thrombus deposition may indicate a lower risk of aneurysm wall rupture. The likely reason for this is that the protection of the posterior aneurysm wall region from ischemia allows it to be better oxygenated and nourished, which is supplied by the lumbar artery from the posterior aorta [55]. Furthermore, both growth differences and biochemical differences in different regions of the AAA drive the hypothesis that the distribution of ILTs in the AAA sac might influence the rate of AAA growth. Compared to the anterior wall, the posterior wall is less affected by the thick ILT layer. Taken together, the study draws conclusions from a biomechanical perspective, but attempts to explain them from a biochemical perspective. The findings of this study can be applied in clinical practice, as common CT scans can identify posterior deposits of asymmetric ILT without any special computer modeling. This may increase the interval between CT scans to reduce medical costs and anxiety for patients. In addition, the deposition degree of asymmetric ILT measured by asymmetrical thrombus deposition index (ATDI) can also be integrated into a multimodal rupture risk model [54].

### 3.3. Hemodynamic Factors

While most studies have attempted to understand how the presence of ILT affects the properties of the aortic wall [56,57], few studies have focused on how the presence of ILT affects AAA luminal hemodynamics. In fact, local hemodynamic factors strongly influence AAA progression and rupture [58]. Endothelial cells on the lumen surface directly sense wall shear stresses (WSS). WSS not only regulates vascular tone, but also drives vascular remodeling [59]. AAA rupture is caused by a series of biodegradations mediated by the interaction of the arterial wall and hemodynamics. Among them, hemodynamics includes WSS and blood pressure [60] Qiu et al. found that hemodynamic factors specifically affect AAA rupture depending on the presence and thickness of the ILT. Recycle flow and low WSS have a dual effect. On the one hand, they induce local rupture (negative effect); on the other hand, they promote the formation of thin ILTs (positive effect). Eccentrically located thick-layered ILT may increase the risk of rupture in small AAAs. The reason is that its location within the sac lumen causes chaotic flow patterns and rapid increases in flow resistance [61].

## 4. The Role of Platelets in AAA Progression

Platelets not only dominate hemostasis, but are also mainly involved in thrombosis and inflammation. Platelet activation, spreading, degranulation, and aggregation lead to the formation of platelet-rich hemostats. AAA is highly associated with platelet activation and hemostasis, which leads to the formation of intraluminal thrombi in most patients [62].

### 4.1. The Correlation between Platelet Activity/Count with AAA Progression

Flow disturbance within the aneurysm can promote platelet activation and aggregation [63], which may contribute to the platelet destruction or consumption and subsequent low platelet count in AAA patients. Platelet activation is involved in AAA pathogenesis via membrane receptors and secreted mediators [64].

Platelets can also be activated by systemic inflammation, and subsequently secrete many anti-inflammatory cytokines [65]. However, it remains unclear whether platelets can prevent the formation of AAA by suppressing the immune-inflammatory response.

Platelets might have a protective effect on AAA. Some early studies disclosed the positive correlation between the low platelet count and aortic aneurysm size/AAA’s poor clinical outcomes [66,67,68,69]. Liu et al. found that platelet transfusion could remarkably decrease the inflammatory cells’ infiltration and MMPs levels and significantly suppress AngII-driven AAA development in mice [70]. However, whether platelet transfusion can be used as a therapy for the repair of rAAA is still controversial [71]. Jones et al. reported that platelet adhesion is mainly restricted to areas composed of normal collagen fibers instead of the abnormal ones [72]. The collagen fibril structure is likely to be normal in the early (compared to advanced) stage of AAA, which may explain why in one report, diminished platelet adhesiveness in AAA remained abnormal even after transfusion of normal blood [73]. The insight from the above report is that the adhesion of transfused platelets to each stage of AAA tissue is different, which in turn affects the protective effect of platelets. Therefore, more studies are required to evaluate the platelet adhesion on collagen at different stages of AAA. However, as we discussed in Section 4.3: The role of platelets in ILT formation, platelets have an important role in ILT formation, and the use of anti-platelet therapies continues to be a matter of debate. Therefore, we need more clinical data to support the efficacy and safety of platelet transfusion.

GPIb is a major glycoprotein on the platelet surface. Glycocalicin, an extramembranous portion of GPIb, was higher in AAA patients than in patients who underwent carotid endarterectomy [74]. TxA2 is produced from arachidonate via the COX-1 pathway in activated platelets. TxA2 inhibitor BM-573 could suppress AAA growth in rats [75]. Chemokines platelet factor 4 (PF4/CXCL4) and RANTES (CCL5) are located within platelet alpha granules. CXCL4 and CCL5 levels increase in AAA patients’ plasma and ILT luminal layers [76], and their plasma levels are positively associated with macrophage recruitment in murine AAA models [77]. Peptide inhibitor MKEY, which inhibits CXCL4-CCL5 heterodimers, can both prevent AAA initiation and suppress AAA progression [78].

### 4.2. The Interaction between Platelets and Other Cells in AAA

Proinflammatory mediators and cytokines released by platelets can recruit inflammatory cells, such as T lymphocytes and macrophages [79]. Studies have shown that inflammatory cells such as macrophages, T lymphocytes, neutrophils [80], and dendritic cells [81] infiltrate the AAA aortic wall. The infiltration of these inflammatory cells is considered a pathological feature of AAA [82]. Liu et al. reported a positive correlation between platelet deposition and inflammatory cell infiltration in both human and mouse AAA samples [83]. In addition, in Section 2.1: ILT structure, we drew conclusions regarding the interaction between platelets and other cells in ILT.

### 4.3. The Role of Platelets in ILT Formation

Biasetti et al. revealed a fluid dynamics-motivated mechanism of platelet activation and deposition in AAAs [38]. In this study, a three-dimensional flow visualization method indicated that sufficient shear-induced platelet activation facilitates ILT formation in the aortic lumen.

However, Hansen et al. suggested that biomechanically mediated platelet activation is unlikely to play an important role in thrombosis and that we should consider other mechanisms, such as biochemically induced platelet activation [84].

The role of platelets in the formation of ILT is depicted in Figure 2 and is discussed in Section 2.1: ILT structure. During early ILT formation, platelets expressing P-selectin on their surface stimulate neutrophils to preferentially accumulate in the luminal layer of the ILT [85]. Neutrophils have a high affinity for fibrin, and after binding to fibrin, constitutive apoptosis occurs [29], and then various inflammatory cytokines, proteases, and metalloproteinases [86], as well as myeloperoxidase and elastase [87], are released.

The constant renewal of the luminal layer of the AAA thrombus involves platelet activation [88]. Platelet activation-induced procoagulant microparticle release and P-selectin overexpression lead to procoagulant activity in the luminal layer of ILT [89].

Platelet activation is involved in the constantly renewed biological activity of ILT, and this concept is of great significance for clinicians to further explore the pathogenesis and preventive treatment of AAA. Antiplatelet therapy would be particularly useful for small aneurysms that have not yet developed significant ILT and do not meet the criteria for surgical treatment [90]. Animal models have shown that platelet activation inhibitors can strongly inhibit aneurysm formation [88,90]. This effect was accompanied by reduced leukocyte infiltration and reduced protease release within the mural thrombus, which in turn reduced elastic fiber degradation and enhanced SMC colonization within the thrombus. However, we need more prospective human studies to verify the beneficial effects of inhibiting platelet activation and aggregation in AAA patients.

### 4.4. Platelet-Related Hemostatic Proteins

Numerous studies have shown that hemostatic proteins are potential biomarkers for risk stratification. The reason is that it is significantly correlated with ILT volume, which is thought to influence AAA growth and rupture [6,11,14,16,91,92].

In the plasma of AAA patients, there are elevated levels of several platelet activation markers, such as microparticles, soluble P-selectin, soluble CD40L, and soluble glycoprotein V, which are released from mural thrombi [88].

Dai et al. found that platelet activation is associated with the progression of AAA [90]. P-selectin exposed on the platelet surface promotes leukocyte adhesion, which means platelet activation is involved in regulating inflammation [93]. P-selectin deficiency attenuates aneurysm formation in the elastase aortic perfusion mouse model, mainly manifested by attenuated inflammatory responses and preserved vessel wall integrity [94].

Plasma soluble P-selectin (sP-selectin) is mainly derived from platelets and is a classic marker of platelet activation [95]. Blann et al. reported elevated levels of sP-selectin in patients with AAA, suggesting increased platelet activation [96]. However, another study contradicts this finding, which may be attributed to the proportion of patients with hypercholesterolaemia and/or treated with a statin [97].

Proteomic analysis of proteins sequestered by the ILT fibrin network revealed ectopic expression of platelet-derived proteins such as cohesin and thrombospondin-1 [98]. The production of these proteins may respond to proteolytic processes in the local thrombus milieu, thereby altering or activating protein function.

Factor Xa (FXa) is a serine protease that catalyzes the proteolytic conversion of prothrombin to active thrombin. Activated platelets are one source of FXa [99]. Therefore, platelets in the thrombus may contribute to increasing FXa concentrations in localized aneurysms [100]. The content of FXa was significantly higher in AAAs with mural thrombi than in AAAs without mural thrombi. Rivaroxaban, an oral FXa inhibitor, modulates inflammation and expression of oxidative stress biomarkers in AAAs with mural thrombi [101].

In an angiotensin II-induced apolipoprotein-deficient mouse AAA model, the size of AAA was positively correlated with the level of FXa and protease activator (PAR-2) [101]. FIIa inhibition alone had no effect on AAA progression when these mice were treated with enoxaparin (FXa/FIIa inhibitor), fondaparinux (FXa inhibitor), or dabigatran (FIIa inhibitor). In contrast, the simultaneous inhibition of FXa and FXa/FIIa can suppress AAA growth, reduce steroid receptor activator levels, reduce PAR-2 and MMP-2 expression, and reduce Smad 2/3 phosphorylation. Mechanistically, FXa/FIIa may control AAA growth by downregulating PAR-2-mediated Smad2/3 signaling and MMP-2 expression. FXa inhibition alone controls AAA growth by reducing ILT formation as well as increasing elastin degradation. Taken together, FXa/FIIa plays an important role in AAA growth and ILT-mediated elastin degeneration; inhibition of FXa/FIIa may be a potential clinical therapy for AAA patients [101].

Von Willebrand factor (vWF) induces platelet aggregation upon binding to platelets. vWF is involved in the formation of platelet-rich thrombi [102]. The secretion of vWF in response to inflammation or endothelial destruction plays a non-negligible role in the formation of ILTs [103].

The accumulation of thrombin at the site of vascular injury is one of the main pathways for the continuous recruitment of platelets to the hemostatic thrombus. Yamazumi et al. suggested that the maximum thickness of ILT was positively correlated with increased thrombin generation [104]. However, another study showed that thrombin generation did not correlate with ILT volume [105].

Several studies in different populations have consistently shown that plasma levels of apoA-IV are inversely associated with cardiovascular disease, and its role in platelet activity and thrombosis is a major cause of heart attack and stroke.

Xu et al. [106] discovered that apoA-IV is a novel ligand of platelet integrin αIIbβ3, which becomes an endogenous inhibitor of platelet aggregation and thrombosis by competing with prothrombotic ligands such as fibrinogen. By mild chronic attenuation of platelet hyperactivity, especially after meals, apoA-IV can slow the chronic progression of atherosclerosis and other inflammatory diseases without compromising hemostasis. They also found that apoA-IV was able to prevent vessel closure at the site of stenosis; apoA-IV was more antithrombotic in vivo than in vitro.

Therefore, we can put forward a reasonable hypothesis: the concentration of ApoA-IV in plasma is inversely proportional to the growth of ILT; ApoA-IV inhibits the formation of ILT by inhibiting platelet activation and aggregation; ApoA-IV competes for binding to platelet surface receptors, making it abundant in deposition in the thrombus, which in turn leads to a decrease in peripheral blood levels; ApoA-IV recombinant protein may reduce mural thrombosis and inhibit arterial wall rupture in ILT model mice (this study is ongoing by our team).

The aforementioned platelet-related hemostatic proteins have not been explored in their relation with ILT activity and have also not yet been applied to AAA clinical diagnosis. Future work will focus on their prognostic value of AAA expansion, rupture risk, and endoleak occurrence in patients with small aneurysms or endovascular grafts.

### 4.5. Anti-Platelet Agents

There is one database revealing that long-term antiplatelet medication confers protection against AAA progression/rupture [107]. In the following, we listed the preclinical and clinical trials of anti-platelet agents in AAA studies; the anti-platelet targets are shown in Figure 3.

#### 4.5.1. Aspirin (Acetylsalicylic Acid, ASA)

The antiplatelet drug aspirin can reduce the growth rate of AAA [108,109,110,111]. The mechanism may be that ASA reduces thrombus and inflammation of the aortic wall and stabilizes the aortic wall [111]. Owens et al. [77] found that ASA reduced abdominal aortic diameter in mice, but this was not significant. ASA also reduced thrombi and protected mice from AAA rupture-caused death. The mechanism may be that ASA can reduce platelet and macrophage infiltration into the vessel wall, decrease platelet-derived cytokines and plasminogen activators, and reduce MMP-2/-9 activity in abdominal aortas. Clinical data from Owen et al. showed that treatment with ASA significantly reduced rupture and dissection in aneurysm patients. Furthermore, Rouer et al. found in a mouse model of AAA that aspirin treatment prior to stent-graft implantation reduced graft failure rates without major hemorrhage [112].

Several animal and clinical trials have consistently found that antiplatelet therapy slows the expansion of moderate-sized AAAs and reduces the rate of dissection or rupture [77,111]. However, other researchers have refuted these conclusions through clinical trials [113,114,115]. A retrospective cohort study by Chen et al. showed that low-dose aspirin did not appear to protect AAA patients from death or adverse events during up to 10 years of follow-up [115]. In addition, preadmission use of low-dose aspirin did not reduce the risk of AAA rupture and even increased 30-day mortality [116]. Abdominal aortic aneurysm thrombi have a unique morphology compared to other thrombi, such as intracoronary thrombi, with a highly inflamed “fibrin-rich” component [117]. The reason for the ineffectiveness of aspirin antiplatelet therapy may be that leukocytes are an important source of coagulation-specific factors, and prothrombotic diathesis in AAA may be mainly mediated by sequestered leukocytes rather than activated platelets [118], which leads to the ineffective antiplatelet therapy in AAA patients [119].

#### 4.5.2. Clopidogrel

Clopidogrel is an inhibitor of P2Y12 activation. In two in vivo studies [77,83], clopidogrel treatment significantly inhibited AAA progression in angiotensin II (Ang II)-infused apolipoprotein E (ApoE)-knockout mice. Mechanisms may be that clopidogrel bisulfate protects the aortic vascular smooth muscle cell layer (but does not prevent rupture); reduces abdominal aortic platelet and macrophage recruitment; prevents aortic MMP activation; inhibits ROS production and inflammatory cytokine expression; and reduces mouse plasma concentrations of platelet factor 4 and components of the plasminogen activation system. Furthermore, Owen et al. [77] found that treatment with clopidogrel bisulfate could significantly reduce rupture and dissection in aneurysm patients.

#### 4.5.3. Ticagrelor

Ticagrelor is a potent antiplatelet drug which belongs to a new chemical class called cyclopentyltriazolo-pyrimidines. It can reversibly bind to the P2Y12 receptor on the platelet surface and block the action of the platelet agonist adenosine diphosphate (ADP) [62,120,121]. The mechanism by which ticagrelor treatment inhibits the progression of AAA may be that it prevents the formation of ILT or inhibits the expansion of established ILT, inhibits the recruitment of platelet-leukocyte aggregates, and prevents platelet aggregation. Another AAA animal model validated the above mechanistic hypothesis. In this animal model, a segment of decellularized guinea pig aorta was transplanted to the aorta of a recipient rat. This model is characterized by inflammatory damage to the transplanted aorta, which results in extracellular matrix breakdown, intraluminal thrombosis, and aortic dilation. The experimental results showed that ticagrelor treatment inhibited platelet activation and significantly reduced thrombus size and aneurysm formation 6 weeks after aortic transplantation. More specifically, ticagrelor inhibits inflammatory cell recruitment in thrombus, preserves aortic wall elastin structure, promotes aortic wall smooth muscle cell colonization, and promotes aortic wall healing [90,122]. However, in a multi-center randomized controlled trial [122], ticagrelor did not reduce the growth of small AAAs (maximum diameter 35–49 mm). In this study, ticagrelor did not show a thrombomodulatory effect and, therefore, it is not determined whether ILT plays an important pathophysiological role in AAA growth. The results of this clinical study suggest that although platelet activation plays an important role in the initial stage of ILT formation, whether platelets activated by ADP binding to the P2Y12 receptor continue to play a specific role after ILT formation deserves further exploration. Therefore, from a clinical point of view, the current ADP receptor antagonists could not be drugs targeting existing ILT.

#### 4.5.4. Abciximab

Abciximab is an anti-GPIIa/IIIb platelet inhibitor. Experiments in animal models have shown that Abciximab reduces aneurysm diameter, thrombus size, and the incidence of experimental aneurysms. The mechanism may be that abciximab protects the elastin network structure, reduces neutrophil recruitment, and increases mesenchymal cell colonization in the arterial wall. Notably, this animal model did not exhibit AAA rupture. However, in some model rats, there are intraluminal thrombi resembling human aneurysms. Furthermore, this model demonstrates that platelets contribute to the progression and rupture of AAA [88].

#### 4.5.5. Others

In one study, transcriptomic profiling of platelets from AAA patients revealed upregulation of olfactory receptors (ORs). OR2L13 overexpression on platelet surface suppresses platelet reactivity in both humans and mice. Selective deletion of the OR2L13 ortholog in AAA murine models accelerated aortic aneurysm growth and rupture. It is the first study to show a therapeutic approach inhibiting biomechanical platelet activation via platelet surface receptors [123].

## 5. The limitations in AAA/ILT Animal Models

Currently, the two most widely used mouse models of abdominal aortic aneurysm are the angiotensin II (AngII) infusion model and the elastase model of aneurysm. Animal models have limitations due to their lack of translational applicability to clinicopathology, such as the lack of spontaneous ILT formation in most aneurysm models. However, in a modified elastase aneurysm model, the researchers added 3-aminopropionitrile fumarate (BAPN, a lysine oxidase inhibitor) to the drinking water of animals, which resulted in spontaneous ILT formation. This model more closely mimics human AAA pathogenesis [124].

In addition, repeated injections of weak pathogens can also overcome the limitations of animal models and induce spontaneous ILT formation because these weak pathogens have ILT biological activities, including platelet activation, hemagglutination, and neutrophil retention.

In addition, in another animal model, although rats did not exhibit AAA rupture, some model rats had intraluminal thrombus similar to human aneurysms [125]. Pathological features of this model include leukocyte infiltration, medial degeneration, and thrombosis, which are typical of AAA pathology in humans [126]. Therefore, we can apply this animal model to study the role of platelets in ILT formation and AAA progression.

## 6. Discussion

The most commonly used parameter for AAA rupture risk assessment is aneurysm diameter. However, some rapidly expanding AAAs may rupture before reaching 5.5 cm in diameter, while 5.5 cm is the well-accepted threshold for endovascular or open repair [127].

In the search for more reliable rupture risk markers, ILT, one of the most studied AAA pathological features, has been shown to be strongly associated with AAA expansion and is present in 75% of AAAs [12].

ILT has been shown to reduce peak wall stress in AAA and thus have a protective effect on the arterial wall, which prevents rupture [9,128,129,130]. However, other studies suggest that ILT may act as an inflammatory foci of proteolytic and enzymatic degeneration of the aortic wall, thereby increasing the risk of rupture [10,50,131,132]. We and others [50] have found that the biomechanically protective properties of ILT are overwhelmed by its biochemically destructive properties in small AAAs.

We need more clinical studies to summarize the effect of ILT (stucture/thickness/volume/spatial distribution) on AAA growth rate and arterial wall rupture in AAAs with different diameters, particularly in small aneurysms. ILT promotes aneurysm wall weakness in small AAAs, leading to aneurysm enlargement, but as aneurysm diameter increases, further accumulation of ILT reduces the hemodynamic impact on the aneurysm wall and thus plays a protective role. However, thickening of the ILT creates local hypoxia, making the hemodynamically relevant vasoprotective effects insufficient to counteract the risk of hypoxia.

Although antiplatelet therapy is the standard of care for most cardiovascular and thrombotic diseases, its use in AAA remains controversial and its efficacy is unclear. Cameron et al. [133] suggested that platelet aggregation and activation may play a protective role in the early stages of aneurysm formation; however, platelet activation can enhance the inflammatory response and secretion of proteolytic enzymes in advanced aneurysms. Platelet activation plays a protective role in early AAA (diameter ≤ 40 mm), which may explain previous clinical findings that antiplatelet treatment of early AAA promotes AAA growth and rupture, while for larger AAAs (diameter 40–49 mm), aspirin treatment inhibits AAA expansion. Therefore, we need to further explore the timing of administration of antiplatelet drugs.

Herein, we make the following hypotheses about the timing of anti-platelet drug administration: First, anti-platelet therapy is most applicable for small AAAs with a diameter of 40–49 mm and with/without the presence of ILT. Second, advanced imaging/physical modeling techniques should be applied to assess thrombus and aneurysm status, and anti-platelet drugs should be applied to the AAAs with destabilized ILT and arterial walls, not the AAAs with nominal aortic diameter growth, progression, and low wall stresses. However, there are three scientific issues that need to be elucidated in the study of anti-platelet therapy for AAA: (1) appropriate ILT animal models; (2) targets for anti-platelet drugs; (3) expanded clinical trials. Thus, tailored preclinical/clinical studies would be required to support the above hypothesis, to address the above issues and to clarify whether monitoring ILT development has any clinical benefit.

## 7. Conclusions

We searched the literature to review the study progress in the potential role of ILT in the pathogenesis of AAA. According to available data, there are limited studies about the anti-platelet therapy for AAA, which needs further clinical trials. This review can provide basic information to direct future researches for developing anti-platelet therapy in ILT-potentiated AAA progression/rupture.

## Figures and Tables

**Figure 1 biomolecules-12-00942-f001:**
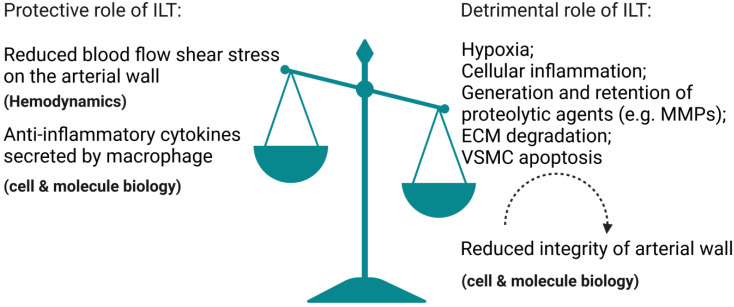
Illustration that demonstrates how the protective biomechanical advantage ILT provides via lowering wall stress is outweighed by weakening of the arterial wall. The illustration elements are from Biorender (https://biorender.com/) (accessed on 30 June 2022; Agreement number: TB243UY73N).

**Figure 2 biomolecules-12-00942-f002:**
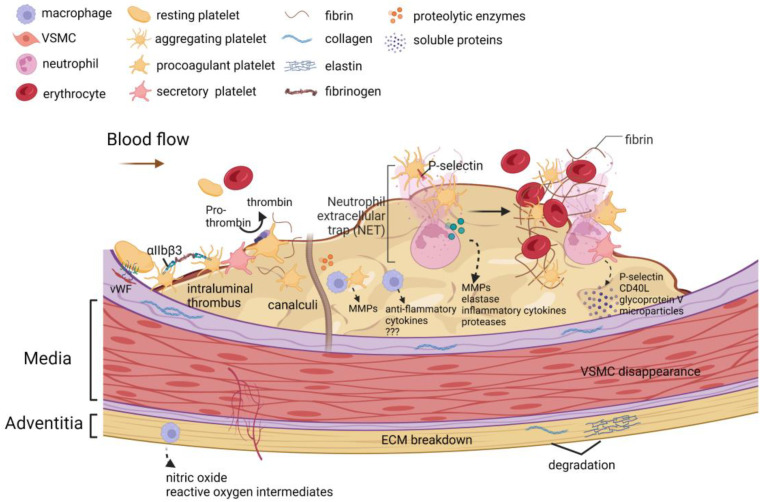
The role of platelets in the formation of an intraluminal thrombus. Platelets play a key role in the formation, expansion, and proteolytic activity of ILTs. Thrombosis is triggered by platelet adhesion to the collagen-vWF complex. Binding of fibrinogen to activated integrin αIIbβ3 triggers platelet aggregation. Procoagulant platelets expose phosphatidylserine on the membrane, which turns prothrombin into thrombin. Secretory platelets in the thrombus release soluble P-selectin, soluble CD40L, soluble glycoprotein V, and platelet-derived microparticles into the circulating blood. During the initial stage of ILT formation, platelet exposure to P-selectin and platelet aggregation stimulates the accumulation and activation of neutrophils preferentially in the luminal layer of the ILT. Neutrophil extracellular traps (NETs) are formed after neutrophil activation. Neutrophils bind to fibrin with high affinity and undergo constitutive apoptosis upon binding, thereby releasing various inflammatory cytokines, proteases, metalloproteinases, elastases, and pro-oxidase (myeloperoxidase). Over time, discrete ILTs mature and form channels called canaliculi. These canaliculi connect the lumen to the luminal layer, which may allow various cell types to infiltrate the ILT. The cell types present in the luminal layer canaliculi are usually degranulated platelets and macrophages. A unique subset of activated macrophages was assembled within the luminal layer. These macrophages secrete various anti-inflammatory cytokines, and these macrophage subtypes are distinct from the macrophage subtypes in adventitia that produce nitric oxide and reactive oxygen species intermediates. Platelets and macrophages are important sources of matrix metalloproteinases (MMPs). The above biological processes promote extracellular matrix breakdown, vascular smooth muscle cell apoptosis, neovascularization, and proteolytic enzyme activation in the vessel wall. The illustration elements are from Biorender (https://biorender.com/) (accessed on 30 June 2022; Agreement number: CI243UXIGU).

**Figure 3 biomolecules-12-00942-f003:**
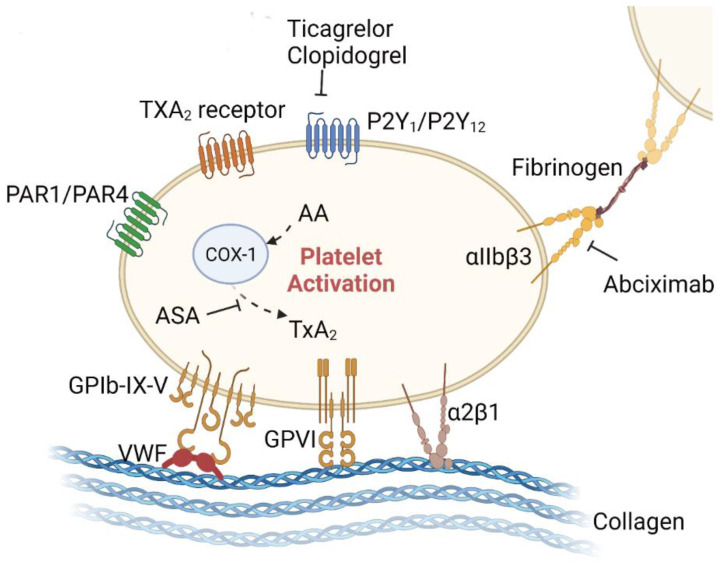
Anti-platelet therapies for AAA in the preclinical stage. Adhesion proteins/receptors on the surface of platelets that can bind to other cells or the extracellular matrix include the integrin family (e.g., α2β1, αIIbβ3), the immunoglobulin superfamily (e.g., GPVI), the leucinerich repeat family (e.g., GPIb-IX-V complex), the G-proteincoupled receptors (PAR-1, PAR-4, P2Y1, P2Y12, and TxA2), and the C-type lectin receptor family (e.g., P-selectin). Some of these receptors are involved in the progression of AAA, while others remain unexplored. Aspirin irreversibly inhibits platelet COX-1, blocks TxA2 production in platelets, and decreases platelet aggregation. P2Y12 receptors are crucial for the platelet activation potentiated by agonists, including ADP, collagen, vWF, and TxA2. P2Y12 receptor antagonists, such as clopidogrel and ticagrelor, inhibit platelet activation by irreversibly binding to P2Y12 receptors and blocking the ADP-dependent pathway. Integrin αIIbβ3 is expressed at high levels in platelets. Upon agonist stimulation, it switches from a low- to high-affinity state for fibrinogen and other ligands, leading to integrin clustering and activating outside-in signaling, which drives platelet aggregation and thrombus consolidation. The illustration elements are from Biorender (https://biorender.com/) (accessed on 30 June 2022; Agreement number: YE243UYYGR).

## Data Availability

Not applicable.

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
