# Peer review of "The Detrimental Role of Intraluminal Thrombus Outweighs Protective Advantage in Abdominal Aortic Aneurysm Pathogenesis: The Implications for the Anti-Platelet Therapy"

_biomolecules, 2022, doi:10.3390/biom12070942_

Round 1

Reviewer 1 Report

I would like to coagulate the authors on an excellent paper. this review is well written and comprehensive.

I would recommend the research team to add a figure which demonstrates the dynamics that occurs within the ILT and how it protects/promotes rupture ( additional to figure 1).

Author Response

Responds to the reviewer’s comments:

Reviewer 1:

I would like to coagulate the authors on an excellent paper. this review is well written and comprehensive.

I would recommend the research team to add a figure which demonstrates the dynamics that occurs within the ILT and how it protects/promotes rupture ( additional to figure 1).

Response:

Thank you for your positive comments.

Whether the ILT thickness/volume can be used to assess rupture remains controversial. The role of ILT in rupture (protective or detrimental) depends on multiple factors (mechanics and molecular biology; some still unclear or unknown). It seems that from a hemodynamic point of view, the ILT reduces the risk of rupture because its attachment reduces the impact of blood flow shear stress on the vessel wall; however, from a cellular and molecular biological point of view, the ILT is associated with arterial wall hypoxia, cellular inflammation and extracellular matrix apoptosis, and can promote aneurysm growth and ultimately lead to aneurysm rupture. The roles of some proteins and cells (eg, macrophages) in the structure of AAA or ILT have not been fully elucidated. In Figure 1 of the original version, we described the role of platelets in the formation of an intraluminal thrombus. This figure also points out regulators in the ILT (from a cellular molecular biology perspective) that regulate arterial wall strength. In summary, through sequestering and activation of platelets, erythrocytes, neutrophils, and macrophages, ILT exposes the vessel wall to a local physiological environment, which is mainly composed of high concentrations of cytokines, proteases, ROS, etc. They collaborate to promote the progression/rupture of AAA. Following your advice, we have drawn a figure which demonstrates the dynamics that occurs within the ILT and how it protects/promotes rupture ( Figure 1 in the revised version).

Reviewer 2 Report

The paper by Ma et al. Entitled “ The detrimental role of intraluminal thrombus outweighs protective advantage in abdominal aortic aneurysm pathogenesis: the implications for the anti-platelet therapy” is a comprehensive review describing the role of intraluminal thrombus on aortic aneurysm development and the use of anti-platelet drugs for early AAA treatment. The authors present one figure where the role of platelets in the formation of an intraluminal thrombus was presented, and one table summarizing the preclinical and clinical trials of anti-platelet agents in AAA studies. The authors also describe the efficacy of currently used anti-platelet drugs on aortic aneurysm progression and rupture in animal and human studies. The review is interesting and aim-orientated. It may be interesting to wide audience. I recommend publishing it in the current form.

Author Response

Thank you for your positive comments.

Reviewer 3 Report

The main topic of this paper is the formation and composition of the intraluminal thrombus (ILT) in abdominal aortic aneurysms (AAA) and its role in the progression of the aneurysm, as well as the platelet activity in the ILT and the possible application of anti-platelet drugs in AAA.

The topic is of great interest, because it is fundamental to understand the pathogenesis of AAA and its progression in details, in order to avoid the rupture, which is a lethal complication in most of cases. The paper is very detailed and clear. There are many studies about the topic, this one tries to comprehend all the knowledge about the intraluminal thrombus, but in the end, it does not give any further information, just some hypotesis that will need further studies to be proven.

The structure of the manuscript needs some changes in order to be more schematic, the paragraphs are too long and have some repetitions that can be avoided.

The abstract is good, it highlights the main topic and the objectives of the review, that are explained in a clear way. The introduction is clear and schematic enough, it would be better to remove rows from 58 to 61, that are a repetition of the objectives already explained in the abstract.

Sub-paragraph 2.1 is not completely clear. It would be better to remove rows from 74 to 77, because the colors of the ILT is not so relevant to understand its molecular composition. It would be better to introduce the topic listing the “protagonists” of the composition, for example citing macrophages, platelets, neutrophils and proteins like P-Selectin, MMP, fibrin and the others all at the beginning of the paragraph, then explaining one by one the role as it is already done. It would be useful also a table in order to schematize the paragraph, listing in a column the cells and the proteins and in the other column their role in the formation/progression of the ILT.

Sub-paragraph 2.2 is well organized and clear.

Paragraph 3 is not so relevant, as the potential protective role of the ILT is already explained in rows 144-145, as well as in the abstract and in the introduction; furthermore the part about the type 2 endoleak is not coherent with an eventual protective role of the ILT in the expansion of AAA. So it would be better to remove the entire paragraph.

Paragraph 4 and its sub-paragraphs are well organized and clear. There are some repetitions that could be removed, for example rows from 212 to 214. The part about the MRI studies is interesting but the clinical application could be difficult, as the MRI is not an accessible exam everywhere.

Sub-paragraph 5.1 is clear, it would be better to add some suggestion about a clinical approach to the results explained in rows from 324 to 332.

The sub-paragraph 5.3 is clear and well organized. The figure and the caption are clear. It would be better to remove the dotted box and to put it as a separate figure in the sub-paragraph 5.5, where the anti-platelets drugs are explained.

In the sub-paragraph 5.4 there are some repetitions that could be removed, for example rows from 401 to 403. Also the part about ApoA-IV is interesting but it would be better to add some suggestion for a clinical use of all these proteins up or down-regulated in patients with AAA and ILT.

In sub-paragraph 5.5 it would be better to remove the table, because the explanation below is absolutely enough and clear, the table appears only as a repetition and does not work as a good scheme. It would be better to add the dotted box in figure 1 as a separate figure (figure 2), with a caption in which is explained the molecular mechanism of every anti-platelet drug considered. In this way, in the sub-paragraph for each drug the explanation of molecular mechanism can be removed. Rows from 481 to 483 can be removed as the topic is not coherent with the progression of AAA correlated to ILT.

The rest of the paper is clear and well organized.

The english is excellent, the phrases are well organized.

The sources are a lot, there is not too much self-citing, some sources may be not so relevant but they add details to the review.

In conclusion, the review is very complete and detailed, it could be more schematic and some repetitions can be removed in order to make the paper more attractive. The topic is interesting and is well explained, but some more suggestions about the clinical prospects may be useful.

Author Response

Reviewer 3:

The main topic of this paper is the formation and composition of the intraluminal thrombus (ILT) in abdominal aortic aneurysms (AAA) and its role in the progression of the aneurysm, as well as the platelet activity in the ILT and the possible application of anti-platelet drugs in AAA.

The topic is of great interest, because it is fundamental to understand the pathogenesis of AAA and its progression in details, in order to avoid the rupture, which is a lethal complication in most of cases. The paper is very detailed and clear. There are many studies about the topic, this one tries to comprehend all the knowledge about the intraluminal thrombus, but in the end, it does not give any further information, just some hypotesis that will need further studies to be proven.

The structure of the manuscript needs some changes in order to be more schematic, the paragraphs are too long and have some repetitions that can be avoided.

  1. The abstract is good, it highlights the main topic and the objectives of the review, that are explained in a clear way. The introduction is clear and schematic enough, it would be better to remove rows from 58 to 61, that are a repetition of the objectives already explained in the abstract.

Response:

We have removed the last paragraph of the Introduction as you suggested.

  1. Sub-paragraph 2.1 is not completely clear. It would be better to remove rows from 74 to 77, because the colors of the ILT is not so relevant to understand its molecular composition. It would be better to introduce the topic listing the “protagonists” of the composition, for example citing macrophages, platelets, neutrophils and proteins like P-Selectin, MMP, fibrin and the others all at the beginning of the paragraph, then explaining one by one the role as it is already done. It would be useful also a table in order to schematize the paragraph, listing in a column the cells and the proteins and in the other column their role in the formation/progression of the ILT.

Response:

We have removed the sentences about ILT colors as you suggested. We introduced the “protagonists” of the composition at the beginning of Sub-paragraph 2.1 and described them one by one. Additionally, we made pruning to avoid duplication.

The following is the revised Sub-paragraph 2.1.

2.1. ILT structure (biochemical perspective)

Most ILTs spatially have three layers, known as luminal, medial and abluminal. There is a clear demarcation and weak adhesion between two adjacent layers. Some specific cells and molecules, such as platelet, macrophage, P-selectin, MMPs, Fibrin, etc., affect the formation and deposition of the ILT.

Over time, discrete ILTs mature and form channels called canaliculi. These canaliculi connect the lumen to the lumen layer, allowing various types of cells in the blood vessel to penetrate into the interior of the ILT from all directions for material exchange [11]. The cell types present in the luminal layer canaliculi are usually degranulated platelets and macrophages. There is a biological balance of coagulation and lysis at the luminal interface between the ILT and circulating blood. Therefore, the lumen of the AAA is rarely obstructed. The physiological activity present in the luminal interface in contact with circulating blood involves platelet activation, which releases microparticles and exposes phospholipids [20]. The involvement of platelets has been elucidated in the following section [5.3. The role of platelets in ILT formation].

Macrophages do not appear to be passively trapped in ILT, as they do not display a necrotic or apoptotic phenotype [11]. A unique subset of activated macrophages was assembled within the luminal layer [21]. These macrophages secrete various anti-inflammatory cytokines, and these macrophage subtypes are distinct from the macrophage subtypes in adventitia that produce nitric oxide and reactive oxygen species (ROS) intermediates [22]. However, the exact contribution of this distinct macrophage subset to the formation of ILTs remains unclear. In the future, we need further studies to determine the distribution and roles of different subtypes of macrophages within the ILT.

P-selectin is an adhesion molecule expressed on the surface of activated cells. At the luminal interface between the ILT and circulating blood, platelet aggregation promotes neutrophil adhesion and activation, a biological process mediated primarily by P-selectin expression [23]. In the most luminal layer of the ILT, activation of neutrophils forms a neutrophil extracellular trap (NET) that traps many enzymes, including proteases, MMP-9, elastase and prooxidases which subsequently are gradually released. The roles of P-selectin and neutrophil have been elucidated in the following section [5.3. The role of platelets in ILT formation & 5.4. Platelet-related hemostatic proteins].

Platelets [24] and macrophages [25] are important sources of matrix metalloproteinases (MMPs), which can proteolytically damage aortic structures. In addition, platelet-derived chemokines can modulate the expression of MMPs from vascular smooth muscle cells (VSMCs) and macrophages [26]. It should be noted that despite the presence of a large number of proteases in the heterogeneous ILT structure, the protease activity mainly resides in the luminal layer [27]. In contrast, the proteases of the abluminal layer were mostly inactive, possibly due to the presence of excess protease inhibitors. Therefore, the balance of protease and its inhibitor expression plays an important role in stabilizing the integrity of the AAA arterial wall.

Some authors suggest that the ILT structure contains large, dense, cell-free fibrin-rich regions that reduce fibrinolysis and stabilize thrombus volume. Fibrin is uniformly deposited throughout the ILT [28]. Fibrin formation results in the retention of plasminogen and tissue plasminogen activator (t-PA), both of which are highly capable of binding fibrin polymers. In AAA, initial retention of plasminogen and t-PA in plasma, urokinase plasminogen activator (uPA) and its inhibitors in neutrophils occurs in the outermost side of the luminal layer near the lumen [29]. Activation of plasmin promotes fibrinolysis, which releases fibrin degradation products. The strongest fibrinolytic activity is at the abluminal interface between the ILT and the wall [30]. In addition to its proteolytic ability, plasmin activates MMPs, mobilizes TGF-β and degrades adherent pericellular proteins such as fibronectin (Fn), ultimately inducing mesenchymal cell detachment and death [31]. The number of neutrophils in ILT is more than ten times higher than in circulating blood because these cells have high affinity for the fibrin-fibronectin network via integrins [32], and bind to P-selectin on platelets [33].

Fn, a dimer of a 250 kDa subunit, is a key component of the extracellular matrix. Fn mainly exists in two forms, known as plasma Fn (pFn) and cellular Fn. Studies have found that pFn can promote platelet aggregation when bound to fibrin, but inhibit platelet aggregation when fibrin is absent [34]. Therefore, we hypothesized that pFn may be endowed with dual functions in AAA (depending on the specific microenvironment); pFn can either support or inhibit ILT formation. We recommend further studies to explore the role of changes in fibrin structure and pFn content in ILT formation and AAA progression.

Based on the above findings, we can conclude that the effect of ILT on the arterial wall is dynamic as ILT grows. Corresponding interventions for different stages of ILT have clinical prospects.

  1. Paragraph 3 is not so relevant, as the potential protective role of the ILT is already explained in rows 144-145, as well as in the abstract and in the introduction; furthermore the part about the type 2 endoleak is not coherent with an eventual protective role of the ILT in the expansion of AAA. So it would be better to remove the entire paragraph.

Response:

We have removed the entire paragraph as you suggested.

  1. Paragraph 4 and its sub-paragraphs are well organized and clear. There are some repetitions that could be removed, for example rows from 212 to 214. The part about the MRI studies is interesting but the clinical application could be difficult, as the MRI is not an accessible exam everywhere.

Response:

We have removed the repetitions and the sentences of MRI studies.

  1. Sub-paragraph 5.1 is clear, it would be better to add some suggestion about a clinical approach to the results explained in rows from 324 to 332.

Response:

Thank you for your suggestion. First, we improved the content of Sub-paragraph 5.1, including supplementing animal model information and adding references to avoid the confusion. We also made recommendations for clinical findings on platelet transfusions to make paragraph sentences more coherent and logical. 

The following is the revised paragraph.

Platelets might have a protective effect on AAA. Some early studies disclosed the positive correlation between the low platelet count and aortic aneurysm size/AAA’s poor clinical outcomes [75-78]. Liu et al found that platelet transfusion could remarkably decrease the inflammatory cells infiltration, MMPs levels and significantly suppress suppress AngII-driven AAA development in mice [79]. However, whether platelet transfusion can be used as a therapy for the repair of rAAA is still controversial [80]. Jones et al. reported that platelet adhesion is mainly restricted to areas composed of normal collagen fibers instead of the abnormal ones [81]. The collagen fibril structure is likely to be normal in the early (compared to advanced) stage of AAA, which may explain why in one report, diminished platelet adhesiveness in AAA remained abnormal even after transfusion of normal blood [82]. The inspiration from the above report is that, the adhesion of transfused platelets to each stage of AAA tissue is different, which in turn affects the protective effect of platelets. Therefore, more studies are required to evaluate the platelet adhesion on collagen at different stages of AAA. However, as we discussed in [5.3. The role of platelets in ILT formation], platelets have an important role in ILT formation and the use of anti-platelet therapies continues to be a matter of debate. Therefore, we need more clinical data to support the efficacy and safety of platelet transfusion.

  1. The sub-paragraph 5.3 is clear and well organized. The figure and the caption are clear. It would be better to remove the dotted box and to put it as a separate figure in the sub-paragraph 5.5, where the anti-platelets drugs are explained.

Response:

Thank you for your suggestion. We have removed the dotted box and put it as a separate figure in the sub-paragraph 5.5, where the anti-platelets drugs are explained.

  1. In the sub-paragraph 5.4 there are some repetitions that could be removed, for example rows from 401 to 403. Also the part about ApoA-IV is interesting but it would be better to add some suggestion for a clinical use of all these proteins up or down-regulated in patients with AAA and ILT.

Response:

Thank you for your suggestion. We are in the research phase regarding the role of ApoA-IV in ILT-potentiated AAA progression. We believe that clinical use of ApoA-IV recombinant protein may reduce mural thrombosis and inhibit arterial wall rupture. Meanwhile, ApoA-IV in peripheral blood can be used as a biomarker to evaluate the risk of ILT-potentiated AAA rupture. As to the clinical use of all these proteins up or down-regulated in patients with AAA and ILT, the direct relation of these plasma markers to mural thrombus activity has never been established, and these markers have never been used as surrogate biomarkers in AAA. The prognostic value of these biomarkers in AAA evolution, enlargement, occurrence of endoleaks, and risk of rupture requires further prospective studies in patients with small aneurysms or with AAAs treated by endovascular graft. We added the above discussion/suggestion to sub-paragraph 5.4.

  1. In sub-paragraph 5.5 it would be better to remove the table, because the explanation below is absolutely enough and clear, the table appears only as a repetition and does not work as a good scheme. It would be better to add the dotted box in figure 1 as a separate figure (figure 2), with a caption in which is explained the molecular mechanism of every anti-platelet drug considered. In this way, in the sub-paragraph for each drug the explanation of molecular mechanism can be removed. Rows from 481 to 483 can be removed as the topic is not coherent with the progression of AAA correlated to ILT.

Response:

Thank you for your suggestion. We have removed the Table 1 as you suggested and also added the dotted box in original Figure 1 as a separate Figure (Figure 3) with a caption in which we explained the molecular mechanism of every anti-platelet drug considered. Besides, we removed the repetitions in sub-paragraph 5.5.

Figure 3. Anti-platelet therapies for AAA in the preclinical stage. Adhesion proteins/receptors on the surface of platelets that can bind to other cells or the extracellular matrix include the integrin family (e.g. α2β1, αIIbβ3), the immunoglobulin superfamily (e.g. GPVI), the leucinerich repeat family (e.g. GPIb-IX-V complex), the G-proteincoupled receptors (PAR-1, PAR-4, P2Y1, P2Y12 and TxA2) and the C-type lectin receptor family (e.g. P-selectin), etc. Some of these receptors are involved in the progression of AAA while others remain to be unexplored. Aspirin irreversibly inhibits platelet COX-1, blocks TxA2 production in platelets and decrease platelet aggregation. P2Y12 receptors are crucial for the platelet activation potentiated by agonists, including ADP, collagen, vWF and TxA2. P2Y12 receptor antagonists, such as clopidogrel and ticagrelor, inhibit platelet activation by irreversibly binding to P2Y12 receptors and blocking the ADP-dependent pathway. Integrin αIIbβ3 is expressed at high levels in platelets. Upon agonist stimulation, it switches from a low- to high-affinity state for fibrinogen and other ligands, leading to integrin clustering and activating outside-in signaling, which drives platelet aggregation and thrombus consolidation. The illustration elements are from Biorender (https://biorender.com/).

  1. The rest of the paper is clear and well organized.The english is excellent, the phrases are well organized. The sources are a lot, there is not too much self-citing, some sources may be not so relevant but they add details to the review. In conclusion, the review is very complete and detailed, it could be more schematic and some repetitions can be removed in order to make the paper more attractive. The topic is interesting and is well explained, but some more suggestions about the clinical prospects may be useful.

Response:

Thank you for your positive comments. We have removed some repetitions throughout the manuscript and rewritten some paragraphs as you suggested in the above comments to make the review paper more schematic, coherent and logical.

Besides, we have checked the manuscript and revised some grammatical and lexical errors, including capitalization, italics, incorrect phrase, word spelling and etc. Also, we highlighted all changes within the document by red-colored text.

Special thanks for your good comments. We appreciate for editors/reviewers’ warm work earnestly, and hope that we have now produced a more balance and better account of our work. Once again, thank you very much for your comments and suggestion.